# A One Health Perspective on Heartworm Disease: Allergy Risk in Owners of Infected Dogs in Gran Canaria (Spain)

**DOI:** 10.3390/ani15213084

**Published:** 2025-10-24

**Authors:** José Alberto Montoya-Alonso, Alfonso Balmori-de la Puente, Noelia Costa-Rodríguez, Jorge Isidoro Matos, Elena Carretón, Rodrigo Morchón

**Affiliations:** 1Zoonotic Diseases and One Health GIR, Biomedical Research Institute of Salamanca (IBSAL), Centre for Environmental Studies and Rural Dynamization (CEADIR), Faculty of Pharmacy, University of Salamanca, 37007 Salamanca, Spain; alberto.montoya@ulpgc.es (J.A.M.-A.); elena.carreton@ulpgc.es (E.C.); rmorgar@usal.es (R.M.); 2Internal Medicine, Faculty of Veterinary Medicine, Research Institute of Biomedical and Health Sciences, University of Las Palmas de Gran Canaria, 35413 Las Palmas de Gran Canaria, Spain; noelia.costa@ulpgc.es (N.C.-R.); jorge.matos@ulpgc.es (J.I.M.)

**Keywords:** *Dirofilaria immitis*, humans, allergic disease, allergy, zoonosis, heartworm

## Abstract

This study provides novel evidence linking *Dirofilaria immitis* infection in dogs to allergy in their owners in a hyperendemic area. While canine heartworm is a well-known veterinary and zoonotic concern, its potential role in triggering human atopy has not been systematically evaluated. Our findings highlight habitat-related risks and suggest that exposure to infected pets may contribute to allergic disease. This study expands the One Health perspective of dirofilariasis and opens new avenues for research on host–parasite–immune interactions.

## 1. Introduction

Heartworm disease, caused by *Dirofilaria immitis*, is a serious mosquito-borne parasitic infection. Once inside the host, the larvae develop into adult worms that reside in the heart, lungs, and associated arteries, leading to pulmonary hypertension, heart failure, and progressive organ damage. Infected dogs may present with clinical signs such as coughing, fatigue, weight loss, and dyspnea, although some remain asymptomatic during the early stages. Without treatment, the disease can be fatal [1]. In humans, pulmonary dirofilariasis occurs when *D. immitis* larvae migrate to the lungs but fail to complete their life cycle. The parasites form pulmonary nodules, which are often detected incidentally on chest radiographs and can be mistaken for neoplastic lesions; however, most cases are asymptomatic, or subclinical [2,3]. Furthermore, there is a significant risk of human exposure in areas where heartworm disease is endemic or hyperendemic [4,5,6].

The island of Gran Canaria (Canary Islands, Spain) is considered a hyperendemic area for heartworm disease and represents a major health problem. The prevalence in dogs has been reported at 16.03%, reflecting a high level of parasite circulation within the canine population. Moreover, the seroprevalence in humans reaches 8.27%, indicating a considerable degree of exposure to infection [7,8]. The risk of infection is not homogeneous across the island. The midland areas, located between the coastal and high mountain regions, where most of the human population resides and where agricultural zones with water storage systems are concentrated, present the highest infection risk and the largest proportion of infected animals [9]. These regions are characterized by dense vegetation, elevated humidity, and warm temperatures, which favor the proliferation of vector mosquitoes essential for the parasite’s life cycle. The climate is typically classified as cold desert, with cold winters and mild to warm summers, or as warm semi-arid, with an average annual temperature above 18 °C.

In this context, the situation in Gran Canaria exemplifies the relevance of the One Health approach, which recognizes that human, animal, and environmental health are interconnected. Understanding heartworm disease under this framework is essential to assess the zoonotic risk and to design integrated control strategies involving veterinary, medical, and environmental disciplines.

In the Canary Islands, where Gran Canaria is located, the prevalence of asthma in humans is 17.2%, well above the Spanish national average of 5.7% [10,11,12]. In general, the prevalence of allergic diseases on the island is high, and the most frequent clinical manifestations are asthma, rhinitis, and eczema, which are considered part of a multifactorial allergic pathology. Several environmental factors may contribute to these conditions, including pollen, dust mites, animal and plant epithelia, or wind transporting particles and dust from Africa. In addition to these environmental triggers, other factors may also play a role in the development of allergic diseases. Previous reports have suggested that contact with *D. immitis* may represent a risk factor for allergy development in humans. In one study, total IgE concentrations were elevated in 34.5% of seropositive human samples and in 8.1% of seronegative samples for IgG against *D. immitis*. Also, the presence of specific IgE against heartworm was detected only in seropositive individuals (17.2%) [13]. The IgE response was mainly directed against two parasite molecules, galectin and aldolase, which may be essential for parasite survival and simultaneously involved in triggering allergic reactions in individuals residing in heartworm-endemic areas [14].

Similar associations between helminthic infections and allergic processes in humans have been reported for *Ascaris lumbricoides*, *Toxocara canis*, *Trichuris trichiura*, and *Schistosoma mansoni*, among others [15,16,17,18,19,20].

With regard to dirofilariasis as a zoonosis, the relationship between the risk of infection and the possible development of allergic reactions in humans living in close contact with infected animals has not yet been fully addressed. However, studies have shown that *D. immitis* proteins are recognized by human IgG antibodies in people living with microfilaremic dogs, providing further evidence of the zoonotic risk [21]. Within the One Health framework, understanding the interaction between canine infections and human immune responses is essential for assessing the broader impact of this parasite on public health. Therefore, the objective of this study was to analyze the association between *D. immitis* infection in dogs and the occurrence of allergic diseases reported by their owners on the island of Gran Canaria, a hyperendemic region for heartworm disease.

## 2. Materials and Methods

### 2.1. Gran Canaria

Gran Canaria is a Spanish island (Canary Islands archipelago) in the Atlantic Ocean (27°37′–29°25′ N, 13°20′–18°10′ W) approximately 150 km off the northwest coast of Africa. It covers an area of about 1560 km^2^, and features a diverse orography dominated by volcanic mountains, deep ravines, and steep coastal cliffs. The island’s subtropical climate, shaped by the northeastern trade winds, creates contrasting microclimates—from humid northern forests to arid southern zones—earning it the description of a “miniature continent” [22]. These environmental gradients, together with high human and canine population densities (869,984 inhabitants and 280,812 registered dogs in 2024) [23,24], make Gran Canaria a suitable setting for vector proliferation and the transmission of zoonotic diseases such as *D. immitis*.

### 2.2. Samples

This study included data from 644 dogs and their respective owners, collected at the Veterinary Teaching Hospital of the University of Las Palmas de Gran Canaria, which represents a medicalized subset of the general canine population, between December 2023 and December 2024 during routine veterinary visits. Eligible dogs were older than 8 months, had no previous history of heartworm infection, and had not received prophylactic treatment for heartworm. For each animal, the following variables were recorded: age, sex, breed, habitat, location, and the result of a rapid antigen detection test for *D. immitis* (Urano Vet^®^, Barcelona, Spain). Human data were obtained from the owners and clients of the dogs tested in this study. Information collected included age, sex, and self-reported history of allergic disease (asthma, rhinitis, eczema), based on anamnesis provided voluntarily by each participant.

Confidentiality and anonymity of both canine and human data were strictly maintained. Informed consent was obtained from each owner in writing; to standardize data collection and minimize interviewer bias, a structured questionnaire was used during the anamnesis. In addition, probing questions were employed when necessary to ensure consistency and completeness in the information gathered from all participants. The study was conducted in accordance with current Spanish and European legislation on animal protection. According to Royal Decree 53/2013, the use of clinical veterinary data for research purposes was used after informed consent from the owners, which was secured prior to inclusion in the study.

### 2.3. Statistical Analysis

Statistical analyses were performed using R software v.4.2.3. [25]. Descriptive statistics for qualitative variables were expressed as frequencies and percentages. Initially, potential factors influencing *D. immitis* positivity in dogs (breed, sex, age, location, and habitat) and the occurrence of allergic disease in owners (age, sex, and location) were analyzed independently. Subsequently, cross-analyses were conducted to further explore associations between canine heartworm infection and owner allergy. The relationships between pairs of categorical variables (e.g., factor–disease, factor–allergy, disease–allergy) were evaluated using Pearson’s χ^2^ test or Fisher’s exact test when expected cell counts were small. For multiple comparisons, Bonferroni correction was applied to adjust *p*-values. Statistical significance was defined as *p* < 0.05, or *p* < 0.0167 in cases of three-group comparisons after Bonferroni adjustment, with subsequently stringent thresholds applied with more groups. The odds ratio (OR) measured the association between factor levels (higher or lower than 1). Normality assumptions were not required in the analysis.

## 3. Results

Results showed that 46.4% (299/644) of dogs tested positive for *D. immitis*, while 43.8% (282/644) of owners reported allergic disease during the anamnesis. Whereas 24.64% of owners of dogs that tested negative for heartworm disease were allergic, 65.89% of owners reported allergic disease when their pets tested positive.

In fact, owners of dogs testing positive for *D. immitis* exhibited a significantly higher prevalence of allergies (~6 times higher based on OR estimation) compared to owners of negative dogs (χ^2^ = 109.05, df = 1, *p* < 0.001) (Figure 1, Table 1). When analyzing positive dogs and allergic owners, no significant associations were observed between owner gender and dog sex (χ^2^ = 0.76, df = 1, *p* = 0.38) or between owner age and dog age (Fisher’s exact test, *p* = 0.70). Among dogs with heartworm, no associations were detected between the dog’s sex and the presence/absence of allergies in their owners (χ^2^ = 0.22, df = 1, *p* = 0.64) or between the owner’s gender and presence/absence of allergies (χ^2^ ≈ 0, df = 1, *p* = 1).

Conversely, significant associations were found between the pet’s habitat and the owner’s allergy status among dogs with heartworm (χ^2^ = 18.13, df = 2, *p* < 0.001). Owners of dogs living in mixed indoor–outdoor habitats showed higher rates of allergy than those of dogs living indoors (~5 times higher based on OR; χ^2^ = 17.96, df = 1, *p* < 0.001) or outdoors (χ^2^ = 7.34, df = 1, *p* = 0.007). No significant differences were observed between the indoor and outdoor groups (χ^2^ = 3.74, df = 1, *p* = 0.053) (Figure 2, Table 2).

Regarding canine variables, habitat was significantly associated with infection (χ^2^ = 8.89, df = 2, *p* = 0.012), with dogs living indoors showing lower infection rates compared to those in outdoor (χ^2^ = 5.72, df = 1, *p* = 0.017; not significant after Bonferroni correction) or mixed environments (χ^2^ = 6.45, df = 1, *p* = 0.011). No significant differences were observed between outdoor and mixed habitats (χ^2^ = 0.28, df = 1, *p* = 0.60). By contrast, neither age (χ^2^ = 7.25, df = 1, *p* = 0.064) nor sex (χ^2^ = 0.003, df = 1, *p* = 0.95) influenced positivity to *D. immitis*. Similarly, location (Fisher’s exact test, *p* = 0.999) and breed (Fisher’s exact test, *p* = 0.034; pairwise comparisons nonsignificant after excluding low-frequency cases) were not associated with infection after excluding low-frequency cases.

With respect to owner-related factors, owner allergy was significantly related to the dog’s habitat (χ^2^ = 13.04, df = 2, *p* = 0.001), with higher rates observed among owners of dogs living outdoors (χ^2^ = 7.07, df = 1, *p* = 0.008) and in mixed indoor-outdoors environments (χ^2^ = 10.69, df = 1, *p* = 0.001) compared to indoor conditions. However, no differences were detected between outdoor and mixed habitats (χ^2^ = 1.01, df = 1, *p* = 0.32) and neither between owner gender and dog sex preference (χ^2^ = 0.22, df = 1, *p* = 0.64) or between owner and dog age (χ^2^ = 6.74, df = 9, *p* = 0.66). Owner age (χ^2^ = 1.46, df = 3, *p* = 0.69) and gender (χ^2^ = 2.31, df = 1, *p* = 0.63) were also not associated with the presence of allergies.

## 4. Discussion

Elevated IgE levels are well-established immunological markers of both parasitic infections and allergic responses [19,26,27,28]. In helminth infections, the immune response is predominantly skewed toward a Th_2_ profile, characterized by the activation of IL-4, IL-5, and IL-13 pathways, increased eosinophil recruitment, and IgE class-switching. This immunological polarization, while fundamental for host defense against multicellular parasites, overlaps significantly with the mechanisms that drive allergic diseases such as asthma, rhinitis and eczema. Consequently, it has been proposed that helminth infections may modulate allergic responses, either exacerbating or, paradoxically, protecting against atopy depending on factors such as parasite species, infection intensity and host immunogenetic background [29].

Previous studies have demonstrated that several nematodes possess allergenic properties capable of inducing hypersensitivity reactions in humans and animals. For example, sensitization to *Anisakis simplex* has been associated with severe allergic reactions, including anaphylaxis [30]. Likewise, infections by *Ascaris* spp., *Toxocara* spp. and *Dirofilaria* spp. have been linked to elevated IgE levels and an increased prevalence of allergic manifestations [31,32,33]. These findings indicate that nematode-derived molecules can act as potent allergens, and structural similarities with common environmental allergens may promote immunological cross-reactivity. It is therefore hypothesized that similar mechanisms could underlie the association observed between *D. immitis* exposure and allergic conditions in humans living in endemic areas.

The mechanistic underpinnings of this association can be contextualized within current immunoparasitological knowledge. Chronic exposure to *D. immitis* antigens, such as galectin and aldolase, has been shown to elicit IgE responses in humans [13]. These molecules are not only crucial for parasite survival and immune evasion but also possess allergenic properties capable of driving mast cell and basophil activation. Comparable mechanisms have been described in other helminth infections, where persistent antigenic stimulation induces immune dysregulation and contributes to allergic disease [34,35,36,37].

In addition, helminths produce a wide array of excretory-secretory products (ESPs), which interact with host immune cells. Some ESPs mimic host proteins or allergens, while others act as adjuvants, skewing the immune system toward Th_2_ polarization. The balance between immunosuppressive and immunostimulatory properties of these molecules determines whether helminth infection protects against allergy (the “hygiene hypothesis”) or enhances allergic reactivity [38]. In the case of *D. immitis*, current evidence supports the latter scenario, at least in endemic areas with continuous antigenic exposure.

Gran Canaria presents a unique epidemiological setting with heartworm transmission deeply influenced by its heterogeneous ecology. The prevalence of *D. immitis* in dogs (16.03%) and seroprevalence in humans (8.27%) are among the highest reported in Europe [7,8]. The island’s environmental characteristics—characterized by warm semi-arid and cold desert microclimates, agricultural midland areas with water storage systems, and stable mosquito populations—create optimal conditions for sustained transmission [9]. These ecological gradients play a key role in shaping exposure risk, not only by supporting high vector density but also by increasing human contact with mosquitoes carrying filarial larvae.

Importantly, dog-related intrinsic factors such as the age and sex of the dogs were not associated with infection status, consistent with previous reports showing that exposure to competent vectors is the main determinant of *D. immitis* transmission [1,39]. This underscores the importance of ecological and habitat-related variables over individual host variables in shaping infection patterns. In this sense, habitat functions as a proxy for vector exposure and, simultaneously, for environmental allergen load, as areas with dense vegetation, agricultural activity, and higher humidity typically harbor both mosquito breeding sites and elevated aeroallergen concentrations.

Regarding the dogs’ habitat and allergic disease reported by their owners, the results revealed a significantly higher percentage of allergies among owners of infected dogs living in mixed habitats. This could be due to increased exposure to *D. immitis* and its antigenic products compared with those in an indoor environment [8,40]. This finding underscores the ecological dimension of the study as one of its major strengths: by incorporating habitat as a key analytical variable, it integrates environmental risk factors into the epidemiological interpretation of both zoonotic transmission and allergic sensitization. Such an approach aligns with a One Health framework, emphasizing the interplay between human, animal, and environmental health. However, the absence of significant differences between outdoor and indoor habitats suggests that additional factors, such as proximity and frequent interaction with dogs, may influence allergy development. It is important to consider that the distribution and prevalence of canine heartworm in the Canary Islands is heterogeneous and strongly influenced by climatic, demographic, and pet management factors in the studied areas.

Finally, while the sex and age of dogs do not appear to influence the likelihood of infection—again consistent with earlier reports that emphasize environmental exposure and vector density as the key drivers of transmission [1,39]—the dogs’ habitat does play a role. Infection positivity was lower in animals living exclusively indoors, reinforcing the role of vectors in disease transmission [41]. Therefore, the consistent role of habitat emphasizes how human decisions regarding pet management directly influence both animal and human health outcomes, underscoring the “One Health” perspective of zoonotic parasitoses.

While occupational and foodborne parasite-allergy associations are well documented, such as allergic reactions to *Anisakis* in fish consumers or urticaria in farmers handling *Bruchus pisorum* [42,43], few studies have addressed allergy development in the context of cohabitation with infected companion animals. This is surprising given the global importance of zoonotic helminths and the close contact between humans and pets. The findings of the present study, therefore, represent a novel contribution to the field, opening new avenues for understanding how domestic reservoirs of zoonotic parasites may influence allergic disease epidemiology in humans.

Moreover, the Canary Islands are notable for having one of the highest asthma prevalences in Spain (17.2% vs. 5.7% national average) [10,11,12]. These high rates of allergic diseases have traditionally been attributed to well-established environmental risk factors, such as high levels of house dust mites, seasonal pollen loads, intrusions of Saharan dust, and urban pollution. However, these factors alone do not fully explain the geographical variability in the prevalence of allergies in the archipelago. Within the ecological context of the island, habitat characteristics that influence vector density and aeroallergen distribution may modulate allergy risk. However, the possible contribution of parasitic infections to this complex picture of environmental allergies has not been systematically investigated. In this ecological setting, exposure to *D. immitis* emerges as a plausible additional factor, raising the hypothesis that parasite-derived antigens and chronic antigenic stimulation could contribute in part to the unusually high prevalence of allergic diseases observed on the archipelago.

From the parasite perspective, the capacity of *D. immitis* to produce a broad array of antigenic molecules, some of which are recognized by human IgG and IgE, makes it a strong candidate for allergy induction [21]. Unlike other helminths that may downregulate allergic responses through immunomodulation, *D. immitis* appears to promote sensitization, especially under chronic exposure scenarios. This observation highlights the need for longitudinal and sero-epidemiological follow-up studies to clarify the temporal relationship between exposure, immune sensitization, and clinical allergy, as well as to elucidate the underlying immunopathological mechanisms.

From a One Health perspective, the implications are notable. Preventing canine heartworm infection through prophylaxis and vector would not only improve animal health and welfare but also reduce environmental parasite circulation and zoonotic risk. Vector density, species composition, and feeding behavior likely differ across habitat categories, influencing both the risk of canine infection and potential human sensitization through shared environmental exposure. At the human level, allergy epidemiology in endemic regions may need to incorporate parasitic exposure as a cofactor, particularly among individuals living in close contact with infected dogs or residing in ecologically favorable habitats for transmission. Furthermore, integrated veterinary–medical surveillance programs could facilitate the early detection of overlapping patterns of *D. immitis* infection and allergic disease. Such coordinated approaches, combining canine screening, vector monitoring, and serological assessments of parasite-specific IgE in humans, may support more effective risk mitigation strategies. Importantly, these interventions offer reciprocal benefits: reducing parasite transmission enhances both public health and animal health, exemplifying the mutual advantages of One Health-driven policies.

Several limitations should be acknowledged. First, the cross-sectional design does not allow causal inference; longitudinal studies are needed to establish temporal relationships between exposure and allergy onset. If data were available on the duration of allergic symptoms experienced by owners in relation to the detection of heartworm infection in their dogs, this could help readers assess the possible temporal association between the two conditions. Behavioral factors such as the frequency of contact between owners and dogs are also not analyzed, which could also influence exposure to *D. immitis* antigens or their vectors. Second, allergy diagnosis relied on self-reported symptoms and medical history rather than standardized clinical testing, which may introduce bias. Additional unmeasured confounders—such as owner lifestyle, occupation, smoking habits, socioeconomic status, and varying levels of exposure to common aeroallergens (e.g., pollen, dust mites, Saharan dust particles)—could also influence the observed associations and were not fully controlled for. In addition, sampling was carried out at the University Veterinary Hospital, which may have introduced a possible selection bias, as the dogs and their owners who attended the hospital may differ from the general population in terms of health awareness, preventive care practices, and socioeconomic status. Third, serological confirmation of *D. immitis* exposure in humans was not performed, which limits the ability to disentangle direct sensitization from indirect exposure effects, or other factors present in the environment, which limits the ability to directly link allergic sensitization to parasite exposure and prevents distinction between true zoonotic transmission and indirect environmental exposure.

Future studies should incorporate human serology and environmental allergen quantification to better disentangle these interactions and should distinguish between allergic processes arising from actual human seropositivity to *D. immitis* antigens and those resulting from indirect exposure to infected pets and their microenvironment. Longitudinal studies combining serological, immunological, and clinical approaches will be essential to confirm causality and to better understand how parasite-derived molecules interact with host immune pathways to modulate allergic responses.

## 5. Conclusions

This study documents an epidemiological association between *D. immitis* infection in dogs and allergy in their human owners in a hyperendemic European region. The results suggest that cohabitation with infected pets may represent a novel risk factor for allergy development. Given the ongoing spread of heartworm disease under climate change and vector expansion, this association may have a global relevance beyond the Canary Islands. A multidisciplinary One Health approach will be essential to fully elucidate the interactions between zoonotic parasites and allergic diseases.

## Figures and Tables

**Figure 1 animals-15-03084-f001:**
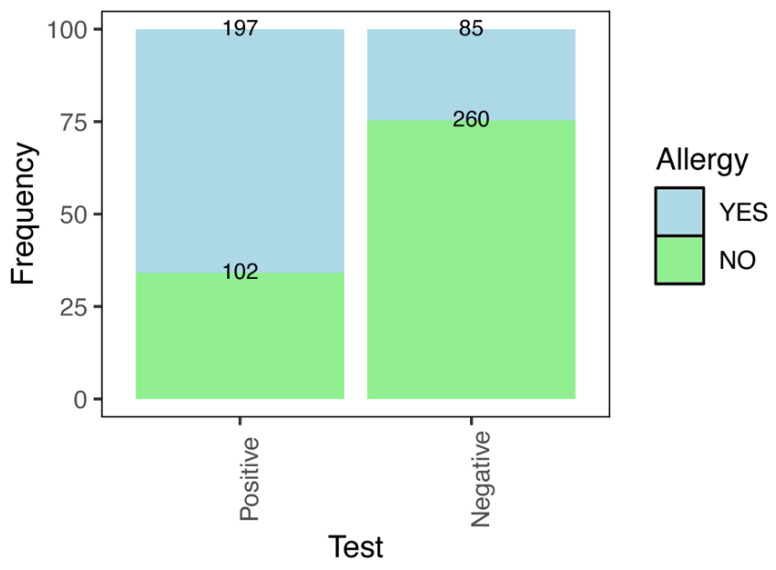
Relative frequency (%) of owners who developed atopy, based on dogs which tested positive and negative in the UranoVet test. Absolute frequencies are represented at the top of the bars.

**Figure 2 animals-15-03084-f002:**
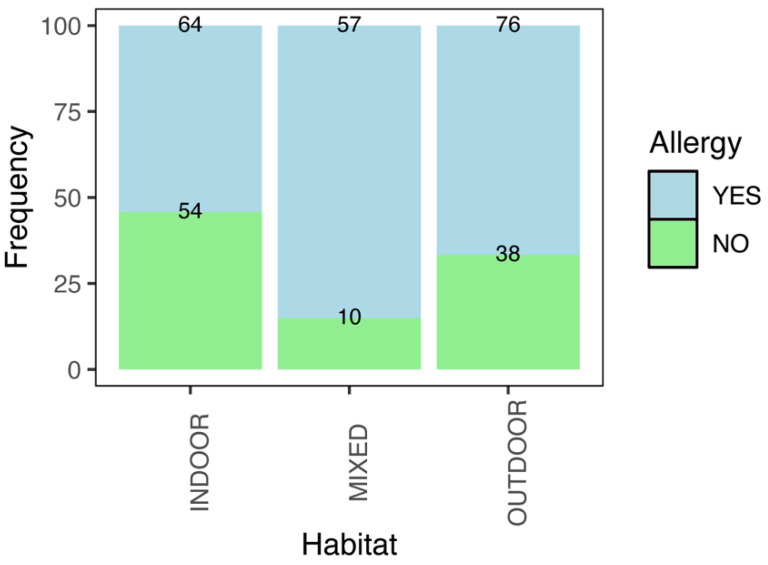
Relative frequency (%) of owners who developed atopy, based on dogs which tested positive in different habitat regimes. Absolute frequencies are represented at the top of the bars.

**Table 1 animals-15-03084-t001:** Absolute and relative frequencies of owners developing atopy based on their pet’s test result. Odds ratio and confidence intervals (CI) are included based on the reference ‘Negative’ test. Ref.: Reference or starting point with which to compare the other variable level.

Heartworm (Test)	Total Cases	Allergy in Owners (Frequency)		Odds Ratio
		Absolute	Relative (%)	(CI 2.5–97.5%)
Negative	345	85	24.64	Ref.
Positive	299	197	65.89	5.91 (4.21–8.35)
Total	644	282	43.79	

**Table 2 animals-15-03084-t002:** Absolute and relative frequencies of owners developing atopy based on their pet’s habitat regime. Note that only positive dogs are shown. Odds ratio and confidence intervals (CI) are included based on the reference ‘Indoor’ habitat. Ref.: Reference or starting point with which to compare the rest of the variable levels.

Habitat of Heartworm	Total Cases	Allergy in Owners (Frequency)		Odds Ratio
Positive Test		Absolute	Relative (%)	(CI 2.5–97.5%)
Indoor	118	64	54.24	Ref.
Outdoor	114	76	66.67	1.69 (0.99–2.89)
Mixed	67	57	85.07	4.81 (2.32–10.82)
Total	299	197	65.89	

## Data Availability

The raw data supporting the conclusions of this article will be provided by the authors upon request. The raw data is confidential and subject to the confidentiality agreement of the University of Salamanca and the company that funded the study.

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
