# Peer review of "A One Health Perspective on Heartworm Disease: Allergy Risk in Owners of Infected Dogs in Gran Canaria (Spain)"

_animals, 2025, doi:10.3390/ani15213084_

Round 1
Reviewer 1 Report
Comments and Suggestions for Authors
This is a good study that should be published to stimulate further research. While there are many confounders which preclude an assessment of causality, I believe that you have provided a good foundation for further investigation. My review, therefore, is mainly around improving study coherence rather than a methodological challenge.

Author Response
Thank you for your comments: Reviewer 1. We have duly considered all of them and believe they have improved the quality of our manuscript. Thank you for your time. For us, this work is very important, as it constitutes the first step in a new, understudied line of research related to parasitic diseases and the allergies derived from them. We have highlighted the replies in red and the texts added to the manuscript in bold.
Reviewer 2 Report
Comments and Suggestions for Authors
The authors present results from an observational study that suggests there is a positive association between infection with heartworm (Dirofilaria immitis) in pet dogs and reported allergic conditions in their owners. The study was conducted in Gran Canaria of the Canary Islands, Spain, where D. immitis is endemic in dogs and humans. The authors explain the study and results well and also do a great job emphasizing limitations of the results in the Discussion.
I have minor suggestions for improvement:
First, I would recommend authors calculate and report odds ratios (OR) and corresponding 95% confidence intervals from the frequency tables used for their analyses. This would provide another measurement of the significance (or lack thereof) of the associations examined, and in particular, an OR would be a formal measurement of risk associated with having allergies in relation to owning a dog infected with heartworm. Such ORs (95% CI) could be presented in Tables 1 and 2.
Second, the absolute frequencies in Figures 1 and 2 are hard to see. I would request that the numbers be displayed in a larger font size.
Author Response
Thank you for your comments: Reviewer 2.
We have duly considered all of them and believe they have improved the quality of our manuscript. Thank you for your time. For us, this work is very important, as it constitutes the first step in a new, understudied line of research related to parasitic diseases and the allergies derived from them. We have highlighted the replies in red and the texts added to the manuscript in bold.
Comments 1: The authors present results from an observational study that suggests there is a positive association between infection with heartworm (Dirofilaria immitis) in pet dogs and reported allergic conditions in their owners. The study was conducted in Gran Canaria of the Canary Islands, Spain, where D. immitis is endemic in dogs and humans. The authors explain the study and results well and also do a great job emphasizing limitations of the results in the Discussion.
Response 1: Thank you for your comments, suggestions and effort to greatly improve the comprehension of the results
Comments 2: I have minor suggestions for improvement:
First, I would recommend authors calculate and report odds ratios (OR) and corresponding 95% confidence intervals from the frequency tables used for their analyses. This would provide another measurement of the significance (or lack thereof) of the associations examined, and in particular, an OR would be a formal measurement of risk associated with having allergies in relation to owning a dog infected with heartworm. Such ORs (95% CI) could be presented in Tables 1 and 2.
Response 2: Thank you for your suggestion. We have calculated the ORs and included their values in their corresponding tables. They are valuable to provide an estimate of risk effect to became allergic when being in close contact with a positive dog (6 times higher).
Comments 3: Second, the absolute frequencies in Figures 1 and 2 are hard to see. I would request that the numbers be displayed in a larger font size.
Response 3: Thank you for your comment. We have displayed histogram values in larger font size so that they are legible
Round 2
Reviewer 1 Report
Comments and Suggestions for Authors
I believe the manuscript is acceptable. Please review for grammatical and punctuation errors. Atopy is still used in a few places. Insure you use allergy or allergic disease throughout.